# Parallel Alterations in Gut and Tumor Microbiota in Pediatric Oncology: Potential Impacts on Disease Progression and Treatment Response

**DOI:** 10.3390/cancers17213426

**Published:** 2025-10-25

**Authors:** Patrik József Szabó, Viktória Sági, Levente Károly Kassai, Renáta Mária Kiss-Miki, Nóra Makra, Dóra Szabó, Miklós Garami

**Affiliations:** 1Pediatric Center, Faculty of Medicine, Semmelweis University, 7-9 Tűzoltó Str., 1094 Budapest, Hungary; 2Faculty of Medicine, Semmelweis University, 26 Üllői Str., 1085 Budapest, Hungary; 3Department of Neurosurgery and Neurointervention, Faculty of Medicine, Semmelweis University, 44 Laky Adolf Str., 1145 Budapest, Hungary; 4Institute of Medical Microbiology, Faculty of Medicine, Semmelweis University, 4 Nagyvárad Square, 1089 Budapest, Hungaryszabo.dora@semmelweis.hu (D.S.)

**Keywords:** pediatric oncology, microbiota, microbiome, tumor microbiota, gut microbiota

## Abstract

**Simple Summary:**

The correlation between the microbiota and tumor initiation and development has been widely studied over the past few years. Several studies have shown that alterations in the gut microbiota can influence the therapeutic efficacy of chemotherapy and immunotherapy against tumors. It was also indicated that the tumorous tissue was not sterile, as it contained various microorganisms. This niche plays a crucial role in tumor formation through different pathological signaling pathways. With this newly discovered knowledge, professionals in the future may focus not only on chemotherapy, immunotherapy, or radiotherapy, but also on restoring the healthy gut microbiota composition to help patients achieve better outcomes.

**Abstract:**

In the last decade, knowledge of gut microbiota has expanded. Several studies have demonstrated a correlation between certain diseases and alterations in gut microbiota. A comprehensive understanding of this complex ecosystem is still lacking; however, this review highlights the importance of microorganisms in oncology. Recently, several studies have demonstrated that the gut microbiota influences therapeutic efficacy and tumor formation, also known as tumorigenesis. We must remember that these microorganisms also play a crucial role in tumor prognosis. Since the discovery of *Fusobacterium nucleatum* in colorectal carcinoma (CRC), it has been established that tumor tissues are not sterile and contain microorganisms that can lead to either beneficial or harmful pathways, affecting tumor size and response to chemotherapeutic agents. Additionally, it should be noted that data on the pediatric population are limited, as this area has not been widely researched due to the low number of cases and the complexity of therapeutic approaches. In children, the only available data are mainly based on hematological malignancies, such as acute lymphoblastic leukemia (ALL). For a better understanding, larger cohorts are required.

## 1. Introduction

### Development of Gut Microbiome

In previous years, it was well known that the development of gut microbiota starts with vertical transmission at birth, when the infant meets and acquires the maternal vaginal and gut microbiota [1]. In comparison to infants born via caesarian section (C-section), their first microorganisms came from the hospital environment, either the maternal skin flora or the employees of the hospital [2]. However, recent studies have shown that colonization occurs earlier than previously thought. Symbiosis begins in utero with the assistance of the placenta [3]. Scientists have declared that the meconium is not sterile and contains a variety of bacteria. With this finding, the previous hypothesis was refuted. Jiménez et al. found that *Staphylococcus epidermidis* was the predominant species in meconium. *Escherichia coli* and *Enterobacter* spp. were detected in most cases, leading to the presence of other predominant groups. The remaining bacteria, *Streptococcus mitis*, *Streptococcus oralis*, *Bifidobacterium bifidum*, *Leuconostoc mesenteroides*, *Rothia mucilaginosa*, and *Klebsiella* spp., were found in only one specimen [3]. After birth, breast milk plays the most significant role in the development of the gut flora. Studies have shown that breastfeeding is crucial for at least 6 months; infants who continue to be exclusively breastfed for less than six months show deficits in weight or length gain from three to seven months or thereafter [4]. Lack of proper breastfeeding also plays a role in reducing the occurrence of acute otitis media, non-specific gastroenteritis, atopic dermatitis, asthma, obesity, type 1 diabetes mellitus (T1DM), type 2 diabetes mellitus (T2DM), leukemia, sudden infant death syndrome (SIDS), and necrotizing enterocolitis (NEC) [5]. Examination of stool samples showed that a lack of proper breastfeeding results in a lower representation of *Bifidobacterium* spp. and a lower level of acetate [6]. A popular alternative feeding method is formula. With formula feeding, special alterations in the gut microbiota have been indicated, with overrepresentation of *Roseburia* spp., *Clostridium* spp., and *Anaerostipes* spp. [7]. Among these children, a special microbial commensal flora was found to play a crucial role in inflammation. As previously mentioned, with lower levels of *Bifidobacterium* spp. and production of acetate, there is a lower activating function of special immune cells, such as T regulatory cells (Treg) [8]. Later, the environment played a major role in the development of flora. Different discoveries have indicated the role of the family in the development of the gut microbiota, especially the correlation between composition and older siblings. Studies have shown that a child without older siblings has a lower α-diversity, which means that, in one sample, there were fewer types of microorganisms in comparison to children with siblings. The relative abundance of *Faecalibacterium* spp. is lower in children without siblings, due to the lower microbiota exposure, which is called the hygiene hypothesis. The microbial exposure helps develop the immune system. It is a well-known fact that *Faecalibacterium* spp. is a commensal anaerobe, which cannot be found in the environment, usually transmitted person-to-person, causing an overrepresentation in siblings; however, this bacterium plays a crucial role in the development of healthy gut microbiota [9,10]. Emerging data regarding the gut microbiota and microbiota in the tumor microenvironment (TME). Microorganisms play a crucial role in tumorigenesis and prognosis. However, it is important to highlight that there are beneficial bacteria that are important for maintaining anti-tumor effects and enhancing chemotherapeutics. *Fusobacterium nucleatum* was the first microorganism to be widely studied. A correlation between colonization and poor prognosis has been proven [11]. On the other hand, the beneficial bacterium *Akkermansia muciniphila* is responsible for a better outcome. This bacterium was enriched in patients with better therapeutic responses to immune checkpoint inhibitors (ICIs). The underlying reason is that *A. muciniphila* enhances the PD-1 response by modulating the tumor microenvironment [12].

## 2. Microbiome and Cancer

### 2.1. Parallel Alterations Between the Gut and Tumor Microbiota in Cancer

The changes that have occurred in the microbiota affect not just the gut microbiota but the tumor microbiota too. When both microbial niches are modified, it is called parallel alteration. The most affected microbiotas are the following: *Fusobacterium nucleatum*, *Escherichia coli*, *Bacteroides fragilis*, *Akkermansia muciniphila*, *Prevotella* spp., *Helicobacter pylori*, and *Malassezia* spp. Some of the important bacteria will be mentioned later, in their respective part of the article. The rest would be elaborated in this section. The *Fusobacterium nucleatum* is enriched not just in fecal samples of colorectal carcinoma patients, but in the tumorous tissue too. They can induce local inflammation via the TLR-4-MYD88-NF-κB pathway. They can also initiate the β-cathenin pathway and induce immune evasion and chemotherapeutic resistance [13,14]. *Escherichia coli* is enriched in the colon microbiota among patients with CRC and hepatocellular carcinoma (HCC). This bacterium could be detectable within the tumorous epithelium and can produce colibactin, which can induce double-stranded DNA (dsDNA) breaks. This can cause genotoxic stress, mutagenesis, and, of course, chromosomal instability [15]. *Bacteroides fragilis* was overrepresented in stool samples of patients with CRC and was also enriched in tumorous tissue and peritumoral mucosa. This bacterium can induce IL-17 production and inflammation and can disturb the E-cadherin pathway [16]. *Akkermansia muciniphila* decreased in advanced cancers, especially in non-responders to ICIs. When it is present in tumorous tissue, it is linked to better T-cell proliferation and advanced therapeutic efficacy [17]. When *Prevotella* species, especially *Prevotella intermedia*, are overrepresented, the TP53 mutation is also elevated [18]. *Helicobacter pylori* was the first microorganism which was described to cause a malignancy. When *H. pylori* is present in the gastric mucosa during a chronic infection, it can cause aberrant DNA-methylation and TP53 mutation. Obviously, the bacterium was available in the gastric microbiota and the tumorous tissue as well [19]. Lastly, the *Malassezia* spp. could be detected not only in pancreatic ductal adenocarcinomas (PDACs), but also in patients’ stool samples. This fungus can activate the complement cascade reaction, with binding to the Mannose-binding lectin (MBL). It can cause tumor-promoting smoldering inflammation [20] (Table 1).

### 2.2. Therapeutic Response

Gut microbiota plays a significant role in various physiological and pathological processes. In oncology, the microbiota can influence the response to different anti-tumor therapies, such as immunotherapy and chemotherapy, as well as tumor formation (carcinogenesis) and disease progression [21]. In previous years, scientists have found that the microbiota plays a significant role in oncological treatments. A study focusing on different malignancies, such as non-small cell lung cancer (NSCLC), renal cell carcinoma (RCC), and melanoma, found that a special bacterium of the commensal flora, *Akkermansia muciniphila*, is responsible for better outcomes in cancer patients [22]. When this bacterium is enriched in patients, they have a better response to ICI treatment than the *Akkermansia*-negative cohort [12]. ICIs are one of the most recent oncological treatments; they focus on eliminating tumors by antagonizing their deactivated immunosurveillance caused by overrepresented PD-1. It is an inhibitory receptor, expressed on T-cells, B-cells, dendritic cells (DCs), and natural killer cells (NK-cells) [23]. When PD-1 binds to its ligand, called PD-L1, its intracellular motifs become phosphorylated, and therefore they recruit a SHP-1/SHP2 phosphatase, which can initiate a downstream signaling among T-cell receptors (TCRs). This can suppress the PI3K/AKT pathway. The other pivotal cells in this process are the B-cells. PD-1 also inhibits the B-cell receptor (BCR) by recruiting the previously mentioned SHP-2, and with this intracellular mechanism leading to a dephosphorylation of fundamental proteins of BCR [24]. Drugs such as nivolumab and atezolizumab focus on reestablishing the physiological balance with receptor–ligand blockade. The exact underlying pathomechanism is as follows. The bacterium enhances anti-PD-1 efficacy by modulating the tumor microenvironment, promoting T-cell infiltration, and reducing immunosuppressive effects [25]. *Akkermansia muciniphila* can increase interleukin (IL) levels, such as IL-12, which is a promoter of T-cell migration and tumoral filtration [26]. In addition to IL-12 elevation, *A. muciniphila* can increase CD8^+^ T-lymphocyte activation [27] and granzyme B and IFN-γ expression, resulting in more efficient tumor elimination [28].

The physiological balance between the M1 and M2 macrophages is important. While M1 macrophages are pivotal in pro-inflammatory states, M2 macrophages are the total opposite; they are responsible for anti-inflammatory effects. M2 plays a fundamental role in oncology. This subtype can produce pro-tumor cytokines, and there is a potential role in accelerating tumor progression [29]. It is important to indicate that the previously mentioned macrophage balance is also shifted, which can cause a polarization toward pro-inflammatory (M1) from anti-inflammatory (M2) macrophages [27] (Figure 1). Apart from therapeutic response, the progression-free survival (PFS) was also better in this cohort. They also reported that patients who had infections and were administered antibiotics had worse outcomes, highlighting the beneficial effects of *A. muciniphila.* The final step of this study was fecal microbiota transplantation (FMT), in which mice that received FMT from responder patients showed better tumor control after PD-1 therapy [12]. However, it is important to note that this study was conducted in a small number of cases. It was observed that supplementation with *A. muciniphila* increased CCR9^+^ and CXCR3^+^ CD4^+^ T-cell recruitment to tumors and enhanced the CD4^+^/Foxp3^+^ ratio [22]. Another component of the commensal flora is associated with better outcomes in cancer immunotherapy. These bacteria were *Bacteroides fragilis* and *Bacteroides thetaiotaomicron*. These two bacteria can alter another pathway of ICIs, the CTLA-4 pathway, in comparison to *Akkermansia*, which can influence the PD1/PD-L1 axis. The underlying elucidation involves the type 1 T-helper cells (Th1) and the previously mentioned IL-12. DCs from the colon can detect bacterial signals, migrate to local lymph nodes, and represent the acquired antigens. With this presentation, DCs can enhance T-cell activation, which synergizes with CTLA-4 blockade to promote tumor elimination [22] (Figure 1). *Bacteroides* spp. are essential for the anti-tumor effects of anti-CTLA-4 therapy. Mice lacking these bacteria showed an impaired therapeutic response compared to other cohorts. They demonstrated the bacterial anti-tumor efficacy of oral administration of *Bacteroides* spp. ICI efficacy was restored after drug administration. The underlying mechanism is based on the IL-12-dependent Th1 response; these microorganisms can enhance the Th1 response, which is crucial for anti-tumor immunity [12]. *Bifidobacterium* spp. has also been shown to affect the therapeutic efficacy. These bacteria are important components of healthy gut microbiota. *Bifidobacteria longum* and *breve* are abundant in healthy individuals [30]. Mice with higher levels of *Bifidobacterium* showed slower tumor growth and improved CD8^+^ T cell infiltration. The main reason why this bacterium is so important is that supplementation with *Bifidobacteria* alone enhances tumor control like anti-PD-L1 therapy. With the combination of *Bifidobacterium* supplementation and anti-PD-L1 antibody treatment, nearly all the malignant cells were eliminated. For a better understanding, the microorganism helps the immune system by enhancing dendritic cells, and through it, there is better and more efficient CD8+ T-cell priming in the tumor microenvironment [31]. The Figure 1 demonstrates how Bifidobacterium, B. thetaiotaomicron, and A. muciniphila modulate immune cell activity and cytokine profiles. Bifidobacterium promotes DC activation, which can stimulate the CD8+ T-cells and enhance IL-13 secretion, leading to an anti-inflammatory response [31]. B. thetaiotaomicron could induce the Th1 cells and enhance the release of IL-12 and IFN-γ [22]. Finally, A. muciniphila could influence the macrophage balance, therefore inducing M1-overrepresentation, leading to a pro-inflammatory state. In comparison, the bacterium could inhibit the anti-inflammatory M2-macrophage subtype [27,28,29].

### 2.3. Tumor Microbiome

The tumor microbiome is a novel concept. The old hypotheses stated that the tumoral tissue is sterile and does not contain microorganisms such as bacteria, fungi, viruses, and protozoa. According to recent data, the microflora of a tumor can initiate direct and indirect immunomodulation [32]. There are clear correlations between bacterial toxins and DNA toxicity [33]. Research on the tumor microbiome began with the Human Microbiome Project (HMP). Researchers have used the 16S rRNA method, combined with whole genome sequencing (WGS). The 16S rRNA sequencing is a sensitive microbiological technique that helps identify bacteria by analyzing the RNA of a specific part of the bacterial ribosome [34]. Short-read shotgun metagenomics can recover now standard community gene catalogs and enable taxonomic profiling, while long-read platforms (PacBio HiFi, Oxford Nanopore) increasingly support contiguous assemblies and recovery of near-complete metagenome-assembled genomes (MAGs), with greater resolution of strain variation and mobile elements [35]. Concurrently, multi-omics workflows—metatranscriptomics, metaproteomics, and metabolomics—are being integrated with metagenomes to connect taxonomy to activity and functional output in situ [36]. For example, increased host-DNA depletion and real-time selective sequencing (adaptive sampling) enhances microbial signal in clinical samples, while Hi-C and improved binning algorithms enable genome bin reconstruction and strain resolution in composite communities more accurately. Finally, the establishment of spatial and single-cell metagenomic/transcriptomic methods has tissue-level resolution and cell-resolved visualization of host–microbe interactions, opening new avenues to investigate tumor-associated microbiota in their microenvironment [37,38].

#### 2.3.1. Gastrointestinal Tumor Microbiome

Colorectal carcinoma is one of the most prevalent malignancies worldwide and a major cause of cancer-related morbidity and mortality. According to the data, CRC accounted for approximately 1.9 million new cases and 935,000 deaths, ranking third in incidence and second in mortality globally [39]. The pathogenesis of CRC is a multistep process involving the accumulation of genetic and epigenetic mutations that potentially drive the transformation of normal colonic epithelium into adenomatous polyps and therefore invasive carcinoma [40]. Key molecular events include inactivation of tumor suppressor genes such as APC, TP53, and SMAD4, as well as activation of oncogenic pathways involving KRAS, PIK3CA, and Wnt signaling [41,42].

Beyond the classical genetic model, emerging data highlight the critical role of environmental and lifestyle factors, such as diet, obesity, and chronic inflammation, in tumorigenesis [43,44]. The gut microbiota has a pivotal role in colorectal carcinogenesis through the production of pro-inflammatory metabolites, toxins, and bile acids that can influence epithelial proliferation and immune responses [45]. Moreover, dysbiosis-induced activation of inflammatory signaling pathways, including NF-κB and STAT3, contributes to tumor initiation and progression [46]. At the molecular level, the tumor microenvironment (TME) plays a crucial role in CRC biology by regulating immune cell infiltration, angiogenesis, and metabolic reprogramming [47].

Recent advances in sequencing and multi-omics analyses have provided new insights into CRC heterogeneity, revealing distinct molecular subtypes with implications for prognosis and therapeutic response [48]. Understanding the complex associations between genetic alterations, environmental exposures, and host–microbiota interactions remain essential to identifying novel biomarkers and therapeutic targets in colorectal cancer. In the international literature, scientists studying colorectal carcinoma (CRC) have discovered that tumor tissue is not sterile but rather contains several microorganisms. Based on this finding, the old hypothesis was tilted. The bacterium found in the tissue was *Fusobacterium nucleatum* with a virulence factor, called FadA adhesin, playing a crucial role in oncogenic pathways, such as the β-catenin pathway. The pathomechanism involves bonding with FadA and E-cadherin, which also induce inflammatory pathways and influence tumor growth [11] (Figure 2). FadA is a specific adhesin protein that is one of the most important virulence factors of the previously mentioned *Fusobacterium nucleatum*. This FadA E-cadherin bond perturbs the E-cadherin-β-cathenin complex. This perturbation leads to a β-cathenin stabilization, and therefore a translocation straight to the nucleus. In the nucleus, the β-cathenin can bind to TCF/LEF and Wnt-target genes, such as c-MYC, Cyclin D1. This pathway regulates stem cell maintenance and proliferation, and protooncogene transcription [49]. The other pivotal adhesin of *Fusobacterium nucleatum* is Fap2. It can bind to an inhibitory receptor called TIGIT, which is located on both NK-cells and T-cells. The intracellular ITIM/ITT-like motifs of the receptors can recruit phosphatases from the SHP family and directly activate effector cells via signaling. These pathomechanistic pathways cause a reduced NK-cell and T-cell cytotoxicity, as well as an IFNγ production in the tumor microenvironment, causing a facilitated immune escape [50,51,52]. *Fusobacterium nucleatum* can activate the Toll-like Receptor 4 (TLR4) on tumor cells and initiate an intracellular signaling by Myeloid Differentiation Primary Response Protein 88 (MyD88). This activation leads to an intracellular cascade featuring the following pathway. MyD88 activates the IRAK, thus activating TRAF6 and finally the NF-κB. NF-κB can transcribe inflammatory and anti-apoptotic signals [53,54]. Studies have also been conducted on PDACs. Scientists have indicated that certain bacterial phyla, such as *Bacteroidetes*, *Firmicutes*, and *Proteobacteria*, are overrepresented in tumor tissues compared with healthy individuals. At the genus level, we can see the overrepresentation of *Pseudomonas* and *Elizabethkingia* in PDAC tissues. In their study, they examined in mice experiments that administering certain bacteria, such as *Enterococcus faecalis*, causes the gut microbiota to be transported to the pancreatic tissue. This finding suggests that there is a possible way in which the gut microbiota can alter the tumor microenvironment, through transportation and creating a special bacterial niche [55,56]. It is important to note that most of the components of the gut microbiota are bacteria, but we cannot forget other species, such as different genera of fungi. In the complex pathogenesis of PDAC, certain fungi, such as *Malassezia* spp., were overrepresented in comparison to the healthy cohort. In the cell wall of *Malassezia* spp., there is a special component, called glycans, which can bind to the MBL and initiate complement cascade activation [57]. When MBL recognizes *Malassezia* surface glycans, it activates the lectin pathway, a subtype of cascade activation. The lectin pathway is mediated by MASP-1/2 proteases. Proteases are responsible for the cleavage of C3 to C3a, thereby activating the fundamental element of cascade activation. Then C3a binds to its receptor called C3aR, which can be found on PDAC cells, promoting their proliferation the oncogenesis [20]. Microorganisms and the consequential cascade activation are responsible not only for tumorigenesis but also for prognosis. C3AR signaling in tumor cells also promotes the epithelial–mesenchymal transformation, which is responsible for metastases and invasiveness [58]. 

Fusobacterium nucleatum’s FadA adhesin interacts with the epithelial cell’s E-cadherin, which causes an activation and nuclear translocation of β-catenin, leading to protooncogenic factor transcription, e.g., MYC and Cyclin D1. This leads to epithelial proliferation. In addition, the F.nucleatum’s lipopolysaccharide and other virulence factors could bind to the TLR-4, which can induce an intracellular signaling via MYD88, IRAK1, and TRAF6, therefore leading to NF-κB activation and transcription of anti-apoptotic and pro-inflammatory genes. If we combine these steps, these cascades establish a tumor-initiating microenvironment via supporting smoldering inflammation [11].

#### 2.3.2. Respiratory Tract Tumor Microbiome

The lower respiratory tract is also thought to be sterile; however, findings show that the lower respiratory tract also has a unique microbial niche [59]. The dominant phyla are *Firmicutes* and *Bacteroidetes*. *Prevotella*, *Veillonella*, and *Streptococcus* were the dominant species at the genus level. These bacteria are the main components of a healthy oral microbiome. Regarding anatomical position, colonization of the lower respiratory tract could originate from the oral flora [60]. Dysbiosis, which is the difference that occurs in the microbiome, appears in 50–70% of cases of respiratory tract malignancies in patients who have suffered from lower respiratory tract cancer and previously had a lower respiratory tract infection [61]. While examining tumorous tissue, they found that severe tumor progression comes with *Thermus* genus overrepresentation, while metastatic lung cancer patient samples were *Legionella*-enriched. These findings may be relevant to the tumor progression and metastatic behavior of certain lung cancer types [62]. Other studies have shown that patients whose saliva samples were enriched in *Veillonella* and *Capnocytophaga* are also diagnosed with lung cancer. This finding could serve as a potential biomarker for the diagnosis of certain tumor types [63]. However, further examinations and larger cohorts are needed to gain a better understanding. Bronchoalveolar lavage (BAL) studies have shown that *Veillonella* and *Megasphaera* species are overrepresented in patients with lower respiratory tract malignancies, compared to healthy individuals [64]. *Veillonella* spp., especially *Veillonella parvula*, is associated with the most dominant taxa during lower respiratory tract dysbiosis. This microorganism is responsible for the upregulation of the PI3K and ERK pathways among patients who are suffering from different lung cancers, such as NSCLC, adenocarcinoma, or squamous cell carcinoma. PI3K activation could be an early event in carcinogenesis [65]. Another fundamental microorganism that plays a role in tumorigenesis is *Legionella pneumophila. L. pneumophila* is associated with Legionnaire’s disease and Pontiac fever. The oncological aspect of this bacterium is known for its capability to induce IL-8 expression. However, this induction was dependent on NF-κB activation. This hypothesis was supported by the administration of NF-κB kinases, which led to decreased levels of IL-8 [66]. Besides interleukins, *L. pneumophila* can induce programmed cell death. This process is caused by the infection of epithelial cells of the alveoli. Programmed cell death is caused by activation of different caspases, such as caspase 3, 8, 9, and 1. It can also trigger the release of High Mobility Group Box 1 (HMGB1). HMBG1 is a damage-associated molecular pattern protein (DAMP). DAMPs stimulate inflammation and recruit immune cells. When HMBG1 is released, NF-κB activation persists. This can lead to persistent production of pro-inflammatory interleukins, such as IL-6, TNF-α, and IL-1β. It can also cause oxidative stress and generate reactive oxygen species (ROS). ROS are known for their DNA-damaging potential. Constant DNA mutations can lead to mutations that can affect crucial protooncogenes and tumor suppressors, such as p53 and KRAS [67,68].

#### 2.3.3. Central Nervous System Tumor Microbiome

The central nervous system (CNS) is one of the most isolated organs of the human body. The blood–brain barrier (BBB) regulates entry into the CNS [69]. According to earlier hypotheses, microorganisms cannot enter the CNS and can only be detected in cases of infections such as meningitis and encephalitis. Evidence supports the existence of a CNS microbiome in healthy individuals. Malignancies may emerge when the optimal balance is disrupted, leading to dysbiosis. One example is IDH-wildtype (isocitrate dehydrogenase) glioblastoma (previously known as glioblastoma multiforme (GBM)). Patients with glioblastoma have a disrupted pro- and anti-inflammatory balance; the crucial participants in this shift could be microorganisms [70]. Certain bacteria of the gut microbiota can produce special fatty acids, such as short-chain fatty acids (SCFAs), which can penetrate through the tight junctions of the BBB and activate the NF-κB pathway in tumor and immune cells. Additionally, certain bacteria metabolize tryptophan to produce metabolites such as indoleacetic acid, indole, and tryptamine, which have immunomodulatory effects [71]. Meningiomas are the most common CNS tumors. Examining meningiomas, scientists have discovered a correlation between tumor formation and gut microbiota composition. When a tumor occurs, there is lower α-diversity. Bacteria such as *Lechnospira*, *Agathobacter*, and *Bifidobacterium* decreased, in comparison to the *Enterobacteriaceae* family, and were overrepresented in this patient population [72]. The decreased level of *Bifidobacterium* can explain tumorigenesis in patients with glioma. A multi-omics-based study has highlighted this correlation. Multi-omics is a complex biological analysis approach that involves genomics, transcriptomics, proteomics, and epigenomics. The ominous bacterium, the *Bifidobacterium*, can inhibit MEK/ERK signaling and suppress Wnt5a mRNA levels, which can lead to suppressed glioma growth. MEK1/2 is a well-known mutation, occurs in patients with glioma. Overrepresentation can indicate a poor prognosis. Supplementing with *Bifidobacterium* spp., e.g., Bifidobacterium *breve*, *Bifidobacterium longum*, *Bifidobacterium lactis*, and *Bifidobacterium bifidum*, can alter the composition of the microbes in glioma, which may indicate a future potential role of this bacterium in oncological therapy [73]. *Lachnospiraceae* can produce SCFAs, such as butyrate, which has an anti-inflammatory and epigenetic effect [74]. Butyrate can inhibit histone deacetylases (HDACs), which can increase histone acetylation, thereby altering gene expression [75].

The most common pathological mutation in glioma patients is the isocitrate dehydrogenase 1/2 enzyme (IDH 1/2), which plays an important role in isocitrate metabolism. This enzyme converts isocitrate into α-ketoglutarate. α-ketoglutarate-dependent enzymes can no longer function normally. Aberrant DNA-methylation has also been observed for abnormal enzymes. An aggressive phenotype is represented by the epigenetic background, which also has a greater chance of being resistant to different treatments. This process can be influenced by a special bacterium called *Brevibacterium* spp., which can alter α-ketoglutarate and glutamate levels [76]. Other studies have shown that patients with gliomas have decreased levels of the *Firmicutes* phylum in their gut microbiota. In comparison, there were increased levels of the *Verrucomicrobiota* phylum and the *Akkermansia* genus. The *Firmicutes/Bacteroidetes* ratio was also altered in this population [77].

#### 2.3.4. Future Implications

Over the last centuries, disease patterns have been shifted from acute infections to chronic diseases, which are metabolic and immune-related, e.g., diabetes, different cancer types, allergies, and neurodegeneration. As the previous sections have dissected, the human microbiota plays a pivotal role in maintaining health and influencing diseases. However, researchers provided new data regarding disease progression and microbial influence, but it is important to note that, to this day, the concrete pathomechanism of chronic diseases remains unknown. Microbial extracellular vesicles (EVs) represent a new era in biological communication and potentially could fill this gap.

EVs are nano-sized, lipid-bilayer particles that could be secreted by both eukaryotic and prokaryotic organisms. They contain different types of proteins, lipids, DNA, RNA, and metabolites that can act as a messenger between cells and organisms. The interesting attribute of these vesicles is they can easily migrate through blood or other body fluids. They could potentially cross barriers, such as the intestinal wall or the BBB. Therefore, they can influence immune regulation, metabolic homeostasis, and inflammatory responses. They can act like an evocator of diseases via triggering systemic inflammation, altering gut permeability, and promoting chronic diseases, e.g., asthma or CRC. In comparison, some EVs could act as a protective agent via promoting mucosal integrity and immune tolerance. Microbial EVs imbalance could precede the dysbiosis among the microbiota.

Last, but not least, EVs could act as a potential biomarker. The composition of the vesicles could be detected in stool, blood, or urine; therefore, they can reflect the current state of the disease. In the future, EVs profiling could serve a non-invasive diagnostic to detect the disease as early as scientifically possible, and it could also be used for disease monitoring [78].

## 3. Pediatric Specificities

Before dissecting pediatric specificities, it is important to note that most adult data are compatible with pediatric studies. The similarity could be based on the fact that some of the pediatric protocols are based on adult schemes. Because of the larger groups of patients, scientists can maneuver the dosage, helping to reach remission among pediatric oncological patients [79]. A systematic review also declares that ICIs used during the treatment of adult melanoma patients are also effective in children with melanoma. The review observed both the PD-1/PD-L1 and CTLA-4 axis [80]. Last, but not least, it is important to highlight that therapeutic agents like sunitinib/pazopanib, used in RCC, could be used among pediatric malignancies [81]. Based on the available international literature, it could be possible that special microorganisms could influence and enhance chemotherapeutic efficacy not only among adult patients but also in the pediatric population.

To the best of our knowledge, there are few publications or data on the correlation between the gut microbiome and tumor microbiome in pediatric solid tumors. The only available information in this population was for patients with hematological malignancies. Acute lymphoblastic leukemia (ALL) is the most common childhood cancer worldwide. In pediatric ALL, there are significant alterations in microbial diversity and composition even at disease onset. *Granulicatella* and *Veillonella* are overrepresented in ALL [82]. Changes in bacterial abundance can be associated with leukemia status. Changes in abundance affect *Granulicatella*, *Carnobacteriaceae*, and *Abiotrophia* [82]. Researchers have examined both the oral and gut flora. In both cases, microbial diversity was reduced and bacterial composition was shifted compared to that in healthy controls [83]. Among the decreased diversity, it was shown that certain bacterial genera, such as *Bacteroides* and *Firmicutes*, were increased in comparison to beneficial genera, such as *Roseburia* and *Faecalibacterium* species, which were reduced during illness [83]. Chemotherapy is another important factor affecting the microbiome. Chemotherapy eliminates tumor cells and destroys healthy cells and microbial agents. It can cause profound changes in the gut microbiota, reducing beneficial bacteria and allowing the overgrowth of harmful bacteria [84]. Well-known side effects such as mucositis, inflammation, and febrile neutropenia are associated with alterations in the healthy microbiome [85,86,87]. Most hematological malignancies in children have a good prognosis. However, it is important to highlight that alterations in the gut microbiota, induced by chemotherapy and antibiotics, have a meaningful impact on their lives in the future. This means that there is long-term dysbiosis in patients who survive ALL, and this can lead to the absence of anti-inflammatory bacteria, such as *Faecalibacterium prausnitzii* [83]. *Facealibacterium prausnitzii* is a butyrate-producing bacterium. Butyrate and other SCFAs are responsible for anti-inflammation and also help strengthen the gut barrier [88]. Restoration of the gut microbiota via fecal microbiota transplantation (FMT) has been widely studied, but currently, there is no available information on this potential therapeutic possibility [83]. Among solid malignancies, retinoblastoma is associated with a potential microorganism-related pathogenesis. Retinoblastoma (RB) is a classic pediatric tumor with a well-known genetic background. Patients with retinoblastoma suffer not just one but at least two genetic mutations. This is called the Knudson-two-hit theory. As Knudson said, every patient has a genetic background, in this case, a mutated RB gene. Patients suffered from another mutation, thus causing the malignancy. Several studies have reported the presence of HPV DNA in RB samples. It is important to indicate that the prevalence of the HPV genome is distributed among patient cohorts [89]. The other tumor type, whose tumorigenesis is strongly associated with bacteria, is nasopharyngeal carcinoma. The Epstein–Barr virus (EBV) is strongly associated with tumors. In a study, EBV-DNA and EBER 1-2 mRNA were found in all specimens [90]. Another study also revealed, with in situ hybridization, that EBV DNA was present in the cytoplasm of neoplastic cells in 9 of 11 samples [91]. Regarding the central nervous system (CNS) tumors, CNS tumors are the second-most common malignancies among children, but the most common solid malignancies. There are also emerging data on the potential correlations between tumors and gut microbiota. Unfortunately, there are no known correlations between tumor microbiota and tumorigenesis in the pediatric population. While characterizing the gut and oral cavity’s microbiota, in patients with solid tumors and healthy individuals, researchers have found that the oral microbiome is enriched in *Veillonellaceae*, *Coriobacteria*, *Atopobiaceae*, *Negativicutes*, and unclassified *Firmicutes* among oncological patients, while among the healthy patients, *Gammaproteobacteria*, *Proteobacteria*, *Burkholderiales*, and *Neisseriaceae* were the most abundant [92]. Sarcomas have also been observed in other solid malignancies. Sarcomas are one of the most common solid malignancies. Patients with osteosarcoma were included in is study. The gut microbiota was examined and compared with serum metabolites. The results indicated an increased abundance of *Alloprevotella* and *Prevotella* genera. Dysbiosis was associated with changes in different pathways, which are related to glycan degradation or the citrate cycle [93].

## 4. Discussion

There are emerging results on the correlation between the gut microbiota and tumor microbiota. The entire field has developed significantly since the Human Microbiome Project (HMP). Several studies have highlighted the potential role of gut microbiota in various diseases and malignancies. Microbial agents such as bacteria, fungi, viruses, and protozoa play crucial roles in tumor formation, progression, and therapeutic response. Certain bacterial species, such as *Akkermansia muciniphila*, assist with ICIs by modifying the PD1/PD-L1 axis. The other axis mentioned earlier was the CTLA-4 pathway. *Bacteroides fragilis* and *Bacteroides thetaiotaomicron* can alter this pathway, which can improve therapeutic efficacy in cancer treatments. One of the earliest tumor types studied was colorectal carcinoma, in which *Fusobacterium nucleatum* was found to influence tumor growth. Other microbiological agents in the tumor microenvironment include *Malassezia* spp. in PDACs, which can bind their glycans to MBL, thereby activating the complement cascade. The lower respiratory tract and central nervous system are traditionally considered sterile and are thought to be free of microbes in healthy individuals. However, recent findings have revealed that microbial components can be present in these areas even in healthy individuals. Disruption of this delicate microbial balance may lead to various diseases, including malignancies. Due to the rarity of childhood cancers, there is limited information on the relationship between gut microbiota and solid tumors. The available data mainly concern childhood hematologic malignancies such as acute lymphoblastic leukemia (ALL). In this population, the pattern observed included alterations in the gut microbiota, such as decreased levels of anti-inflammatory bacteria like *Faecalibacterium prausnitzii and Roseburia* spp. and an increased presence of *Ruminococci* and *Firmicutes*. These changes are associated with worse prognosis and more frequent side effects, including mucositis, inflammation, and febrile neutropenia. As therapeutic agents used in adulthood are also available for children, it is important to highlight the significance of the microbiota. This article details how different microorganisms, such as *Akkermansia*, could alter the therapeutic efficacy of ICI treatment in adults. In future studies, pediatric cohorts should also discuss how these bacteria could influence therapy in children. With the widely spread information of the gut and tumor microbiota, we could design a personalized oncological treatment for children, achieving a better therapeutic response, and even the microorganisms could decrease or eliminate the side effects of these pharmaceutical agents.

The intricate relationship between the gut and tumor microbiota represents a promising area of research with significant clinical implications. However, it is essential to acknowledge that various pitfalls exist in microbiota-related research. Collecting sterile samples is challenging. It is well known that contamination is common; even highly precise researchers cannot eliminate it. Therefore, readers should be cautious when interpreting data from these studies. Another problem that should not be ignored is that, among these studies, the cohort sizes were small. The results can be misleading, or the correlations may not be significant. As it is an emerging branch of medical research, in the future, there is a possibility that we could gather more experience, leading to even larger cohorts or new significant correlations. However, methodological difficulties should not be ignored. Each researcher and research group used different sequencing protocols. This complicates the outcomes. In the future, researchers should use a standardized protocol to avoid this heterogeneity.

Although emerging studies have hypothesized characteristic microbial signatures associated with colorectal cancer, the accuracy of a single microbiota profile as a biomarker remains highly uncertain. The gut- and tumor-associated microbiota are changing in a dynamic pattern and are influenced by multiple factors, including diet, host genetics, disease stage, treatment, and geographic background. Moreover, the underlying individual and intratumoral heterogeneity complicates the identification of universally persistent bacteria and bacterial markers. Current findings are primarily coming from small-scale or region-specific cohorts, and variations in sample processing and sequencing platforms could also possibly limit the results in studies. While microbiota profiling holds great promise for early detection and prognostic stratification, its translation into a clinically accurate biomarker requires large, standardized, and longitudinal studies integrating multi-omics approaches and functional validation.

When examining the tumor microbiome can provide important information about cancer biology and potential therapeutic targets, there are several important limitations to consider. First, the field is still early in its development, and many of the studies are heterogeneous in their methods, sample types, and patient populations, making it difficult to generalize conclusions and compare results among studies. Second, microbial communities are complex, and the host also has many factors that can affect the interpretation of any association, including genetics, immune status, and treatment history. Third, there are technical limitations related to low microbial biomass in human samples, plus potential sample contamination, and while sequencing and bioinformatics approaches have improved, they are often still not consistent across studies. Finally, many studies are observational in nature, so it may also be difficult to establish causal relationships between specific microbes and tumor behavior. Understanding these limitations and the context of findings is important for the interpretation of the microbiome in tumor biology and setting the future research agenda towards better methodologies and standardization.

The emerging field of medical microbiology, focusing on the microbiomes of different organs, is a controversial topic. The main debate among researchers centers on the central nervous system (CNS). However, discoveries have indicated that microbiota exist in a healthy CNS; there are still opponents, so this finding awaits broad scientific validation. Another controversy involves tumor microbiota. In fact, microorganisms are present in tumorous tissue, but there is no clear methodology for distinguishing between the natural microbiota and the microbiota acquired by tumors. Finally, antibiotics have been widely discussed for their ability to alter gut microbiota. However, it is important to note that other factors, such as hospitalization and length of stay, also influence the microbiome, with longer stays increasing the chance for commensal flora to be altered. Diet is also significant, but standardizing diets is challenging due to individual nutritional needs. Although many questions remain in this developing field, current evidence strongly suggests that maintaining microbial balance is vital for health. Dysbiosis can lead to serious health problems, including chronic diseases and cancers. Future clinical practice should include microbiota management strategies to restore microbial harmony, improve treatment outcomes, and enhance patients’ quality of life.

In pediatric oncology, further research is vital to explore not only hematological malignancies but also CNS tumors and other solid cancers. Pediatric microbiota should be studied in the future. There are clear correlations between alterations in the gut microbiota and disease progression. Therefore, it is important to conduct longitudinal microbiome studies among children, and we could gather a lot of information about the development of healthy and perturbed gut microbiota, as well as how a chronic disease could disturb the microbiomics balance or how modern medication could alter the microorganisms in the gut.

In future studies, researchers should focus not only on identifying microorganisms (e.g., bacteria and fungi) but also on characterizing the metabolites they produce. Building on this review’s emphasis on butyrate—a prototypical short-chain fatty acid—future work should systematically evaluate additional metabolites, such as propionate and indole-3-acetic acid (IAA), to clarify their mechanistic roles and clinical relevance for disease progression, therapeutic modification, and the development of prognostic or predictive biomarkers. These directions support the longer-term prospect of microbiota-informed, personalized oncology that integrates tumor-resident and gut microbiota features with modifiable factors such as diet. Microbiome-directed interventions—including dietary strategies and targeted supplementation with defined bacterial or fungal consortia—warrant rigorous, contamination-aware, controlled trials to establish safety, efficacy, and patient-selection criteria. Given the heightened vulnerability of the developing microbiota in infancy and childhood, careful consideration and well-designed early-life interventions, where appropriate, may yield substantial long-term benefits.

## 5. Conclusions

Gut and tumor microbiota hold promise for oncology but require cautious interpretation. Contamination, small cohorts, and heterogeneous methods remain major pitfalls; standardized, contamination-aware protocols are essential. Key confounders—antibiotics, hospitalization, and diet—must be rigorously measured and controlled. Beyond taxa, metabolites (e.g., propionate and indole-3-acetic acid) should be evaluated as mechanistic drivers and predictive or prognostic biomarkers. Ultimately, microbiota-informed and diet-responsive personalized oncology, tested in robust trials, may benefit patients—especially children.

## Figures and Tables

**Figure 1 cancers-17-03426-f001:**
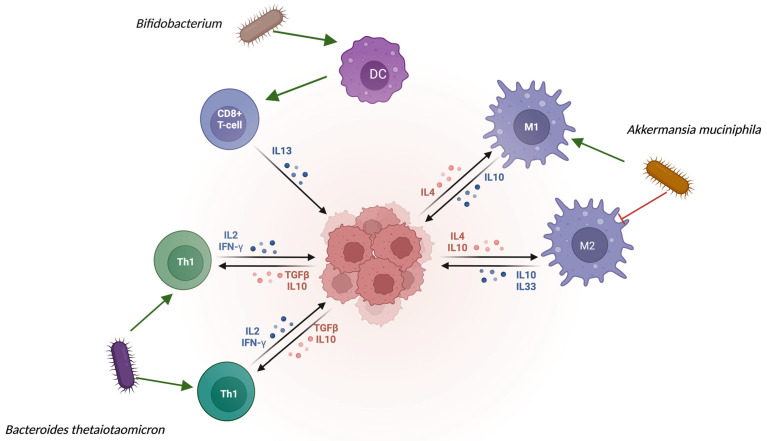
Interactions between gut microbiota and immune cells via cytokine signaling.

**Figure 2 cancers-17-03426-f002:**
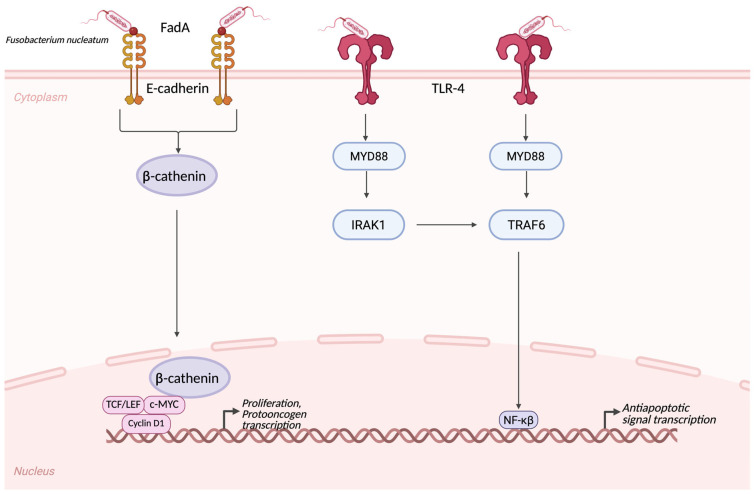
Fusobacterium nucleatum-induced pro-inflammatory and oncogenic signaling in colorectal epithelial cells.

**Table 1 cancers-17-03426-t001:** Most common parallel alterations both in the tumor and in the gut microbiota.

Microorganism	Change in the Gut Microbiota	Alteration in the Tumor Microbiota	Biological Role
*Fusobacterium nucleatum*	Increased in fecal samples of CRC patients	Strong overrepresentation in tumorous tissue, local invasion to the tumor stroma	Induces local inflammation via TLR-4-MYD88-NFκB pathway; induces β-catenin activation [13].
*Escherichia coli*	Elevated in patients of CRC and HCC	It can be found within the tumor epithelium	Colibactin can cause a dsDNA break, and therefore genotoxic stress, mutagenesis, and chromosomal instability [15].
*Bacteroides fragilis*	Overrepresented among patients with CRC	Present in tumorous tissue and in the peritumoral mucosa	Induces IL-17 inflammation, disrupts E-cadherin [16].
*Akkermansia muciniphila*	Decreased among cancer patients, especially in ICI non-responders	Rarely found in the tumorous tissue, but it can alter T-cell infiltration	It can enhance the anti-tumoral efficacy [17].
*Prevotella intermedia*	Increased in feces of GI cancer patients	Could be found in the tumorous tissue occasionally	It is common to be found with TP53 mutation [18].
*Helicobacter pylori*	Can be present in the gastric mucosa as part of a chronic infection	Dominant species in gastric tumors	Induces DNA-methylation and TP53 mutation, and causes a chronic smoldering inflammation [19].
*Malessezia* spp.	It can be found in feces of PDAC patients	Can also be found in tumorous tissue	Activates the complement cascade via the mannose-MBL binding. It can cause a tumor-promoting inflammation [20].

## Data Availability

The original contributions presented in this study are included in the article. Further inquiries can be directed to the corresponding author(s).

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
