# Peer review of "Parallel Alterations in Gut and Tumor Microbiota in Pediatric Oncology: Potential Impacts on Disease Progression and Treatment Response"

_cancers, 2025, doi:10.3390/cancers17213426_

Round 1
Reviewer 1 Report
Comments and Suggestions for Authors
The review is on the gut microbiota and cancer interaction in children. The topic is of interest and novelty, especially considering the lack of relevant pediatric data. The review is below the standard of a high-impact review, however. It is more descriptive than understanding mechanistically, and it fails to critically evaluate the literature. In addition, the numbers and tables are shallow and do not offer mechanistic or functional information that will make a difference to the reader. A thorough revision should be done before the manuscript is ready for publication.
Serious Criticisms
1. Deficiency of Mechanistic Depth
The review just duplicates observational findings without going into mechanistic bases where microbiota play a function in oncogenesis, immune modification, and response to therapy. For instance: Akkermansia muciniphila has been referred to as advantageous in PD-1 response but is not mentioned at all in terms of immunological mechanisms (e.g., T cell recruitment, antigen presentation, epigenetic control by SCFAs).
The story of Fusobacterium nucleatum describes FadA adhesin and β-catenin activation but does not describe usual areas like modulation of tumor-infiltrating immune cells, induction of chemoresistance, and DNA damage.
Lacking a mechanistic framework, the review is closer to a descriptive summary than to an academy-worthy synthesis.
2. Meager Treatment of Pediatric Oncology
The pediatric oncology emphasis, the avowed purpose of the review, is thin and undernourished.
Acute lymphoblastic leukemia (ALL) is treated extensively; solid tumors (CNS, sarcomas, etc.) are not discussed.
The special properties of the growing microbiome of children and their translation into treatment (diet, antibiotic exposure, immune maturation) are not value added.
3. Tables and Figures Are Weak
Table 1: A list of bacterial phyla and genera is given without functional annotation or description. This is useless to readers — why are these taxa of interest in pediatric oncology? What functions or metabolites are they associated with?
Figures: Legends are not explanatory and do not clarify pathways or interactions at the molecular level. For example, the figure of Fusobacterium nucleatum must clarify specifically how FadA–E-cadherin interaction activates β-catenin and inflammatory signaling cascades, instead of saying "influences tumor growth."
Briefly, the visuals are not explanatory and do not function as mechanistic or conceptual overviews.
4. Lack of Critical Analysis
The review does not mention methodological pitfalls in tumor microbiome studies, e.g., risk of contamination, small cohort size, heterogeneity of sequencing protocols, or challenges in the discrimination between tumor-resident vs. translocated microbivalues.
The refuted notion of a "CNS microbiome" is stated as fact, without a mention that it is still highly controversial.
The influence of antibiotics on microbiota is recognized, but confounders (hospital conditions, diet, chemotherapy per se) are not fully considered.
5. Lost Opportunities for Integration
No mention of how microbiota-derived metabolites (e.g., SCFAs, tryptophan catabolites, bile acids) fit into signaling, although this is important to host–microbe communication in cancer.
No mention of immunological circuits (Tregs, Th17, CD8⁺ T cell priming) which link microbiota alterations to treatment outcome.
No theory to explain "parallel changes in gut and tumor microbiota" — the title leads one to expect this, but the article does not provide it.
Minor Concerns
1. Several grammatical mistakes and unnatural sentences (e.g., "patients a be er outcomes") must be worked through carefully.
2. Some less up-to-date or weaker references employed for the arguments presented; more recent 2022–2024 research on microbiota and pediatrics oncology must be included.
3. The "Conclusions" section is too vague and does not list specific research priorities (e.g., the need for longitudinal pediatric microbiome studies, microbiota-modulating interventions, incorporation of metabolomics).
Cumulatively, the manuscript needs to be improved in three ways:
1.
Mechanistic view: Connect individual microbes with molecular and immunological mechanisms of cancer biology.
2.
Pediatric emphasis: Broaden beyond ALL, include pediatric solid tumors, and take developmental microbiome characteristics into account.
3.
Figures/Tables: Substitute descriptive lists with mechanistic or integrative schematics and functional tables.
Until such a point as they are addressed, the manuscript is still too descriptive and fails to attain the standard of analytical sophistication appropriate to a review article within this journal.
Reviewer 2 Report
Comments and Suggestions for Authors
This manuscript presents a comprehensive and insightful review of the emerging relationship between the gut and tumor microbiota in pediatric oncology. The authors have successfully synthesized a wide range of recent studies, highlighting the potential clinical relevance of microbiome alterations in cancer development and therapeutic response. The review is well-structured, informative, and relevant to current trends in oncology and microbiome research.
- The introduction presents many microbiota facts but lacks a focused transition to oncology relevance.
- The manuscript merges pediatric oncology with adult data (e.g., NSCLC, RCC, melanoma). Clarify how adult studies are relevant to pediatric populations.
- The term “parallel alterations” is not sufficiently demonstrated. Recommend adding a figure or table directly comparing gut vs tumor microbiota alterations side by side.
- Table 1: The phyla list is incomplete and inconsistent (e.g., “Bacteriodetes” should be Bacteroidetes; “Bacteriodaceae” → Bacteroidaceae). Correct taxonomy and format in italics per scientific convention.
- The discussion largely repeats results. Expand on mechanistic implications—how microbiota–immune interactions could guide therapy design in children.
- In Conclusions, suggest including 1–2 sentences on how microbiome profiling might integrate into personalized oncology (e.g., microbiome-based biomarkers for therapy selection).
Reviewer 3 Report
Comments and Suggestions for Authors
This manuscript presents an interesting and well-organized review focusing on pediatric oncology in relation to the microbiome. The topic is timely and relevant; however, the current version requires major revisions before it can be considered for publication. My detailed comments are as follows:
-
Line 76 (Introduction): The sentence beginning with “of the relative abundance ~” requires grammatical and contextual correction for clarity.
-
Table 1 (Introduction): Please provide appropriate references for the data presented in this table.
-
Introduction: Including a more comprehensive discussion on the relationship between microbiome composition and disease/health status would strengthen the contextual background.
-
Section: Microbiome and Cancer: Recently, bacterial extracellular vesicles (EVs) have been recognized as key mediators in cancer therapeutics and diagnostics. It is highly recommended to include this aspect in the review.
-
Ref: “A new horizon of precision medicine: combination of the microbiome and extracellular vesicles,” Experimental & Molecular Medicine, 2022.
-
-
Figure 1: The figure lacks detailed explanation of its individual components. Please add more descriptive captions and supplement the related content in the main text for better understanding.
-
Section 2.2 (Tumor Microbiome): Lines 127–131 discuss sequencing technologies; however, this part should be updated to reflect recent advancements since the Human Microbiome Project (HMP), including current trends in sequencing approaches.
-
Section 2 (Microbiome and Cancer): Adding a review on colorectal cancer would further enhance the comprehensiveness and quality of the manuscript.
-
Overall: Abbreviations are inconsistently described across sections. This gives the impression that the manuscript may have been partially edited using AI tools without final proofreading. A thorough revision for consistency and readability is strongly advised.
Reviewer 4 Report
Comments and Suggestions for Authors
The manuscript presented by the authors is an impressive and comprehensive review that sheds light on the association between gut and tumor microbiota in oncology. The detailed information provided in this review will guide clinicians, physicians, and researchers in developing diagnostic and therapeutic strategies, as well as informing treatment approaches.
However, several modifications and revisions are necessary to enhance the quality and clarity of the manuscript.
- The authors have presented the manuscript under the title, “Parallel Alterations in Gut and Tumor Microbiota in Pediatric Oncology: Potential Impacts on Disease Progression and Treatment Response.” Nevertheless, the manuscript lacks a succinct interpretation about “Gut and Tumor Microbiota in Pediatric Oncology” in the abstract. Additionally, the section titled “3 Pediatric Specificities” should be more thoroughly elaborated to justify the relevance of the review's title. It is also recommended that a representative diagram be included in this section to further support the manuscript's focus.
- The figures should be clearly labeled, and their components briefly explained. Furthermore, modifications are needed to elucidate fundamental cancer signaling cascades related to proinflammatory responses induced by the microbiota as well as toxin release. In addition, more detailed and self-explanatory figure legends should be provided for both figures.
- The authors mention the current limitations in available data and studies on this topic. However, the discussion suggests that a single microbiota profile could serve as a potential biomarker, referencing certain previously published studies. Given the complex nature of microbiota within the tumor microenvironment, advocating microbiota as a reliable biomarker requires more robust evidence. A brief explanation clarifying this issue is warranted.
- The authors are encouraged to include a section outlining the disadvantages or limitations associated with reviewing such a complex and insufficiently explored topic.
- What does mean by “Section 6. Patents; Line 267”? Is this a simple author's contribution section? Please clarify.
Round 2
Reviewer 1 Report
Comments and Suggestions for Authors
The authors addressed all my concerns. Please check the abbreviation around the text; for example, short-chain fatty acids (SCFAs) are noted twice.
Author Response
Comment: The authors addressed all my concerns. Please check the abbreviation around the text; for example, short-chain fatty acids (SCFAs) are noted twice.
Response: We appreciate the reviewer's comment. We corrected the abbreviations.
Action: Change has been made in line: 373
Reviewer 3 Report
Comments and Suggestions for Authors
The authors have adequately addressed the revision comments, and the manuscript is now suitable for publication.
Author Response
Comment: The authors have adequately addressed the revision comments, and the manuscript is now suitable for publication.
Response: Thank you so uch for your comment and help. We appreciate it.